# Infusion: Preventing Customized Text-to-Image Diffusion from Overfitting

## Weili Zeng
MoE Key Lab of Artificial Intelligence, AI Institute, Shanghai Jiao Tong University
Shanghai, China
zwl666@sjtu.edu.cn

## Yichao Yan*
MoE Key Lab of Artificial Intelligence, AI Institute, Shanghai Jiao Tong University
Shanghai, China
yanyichao@sjtu.edu.cn

## Qi Zhu
MoE Key Lab of Artificial Intelligence, AI Institute, Shanghai Jiao Tong University
Shanghai, China
GeorgeZhu@sjtu.edu.cn

## Zhuo Chen
MoE Key Lab of Artificial Intelligence, AI Institute, Shanghai Jiao Tong University
Shanghai, China
ningci5252@sjtu.edu.cn

## Pengzhi Chu
Student Innovation Center, Shanghai Jiao Tong University
Shanghai, China
pzchu@sjtu.edu.cn

## Weiming Zhao
Student Innovation Center, Shanghai Jiao Tong University
Shanghai, China
weiming.zhao@sjtu.edu.cn

## Xiaokang Yang
MoE Key Lab of Artificial Intelligence, AI Institute, Shanghai Jiao Tong University
Shanghai, China
xkyang@sjtu.edu.cn

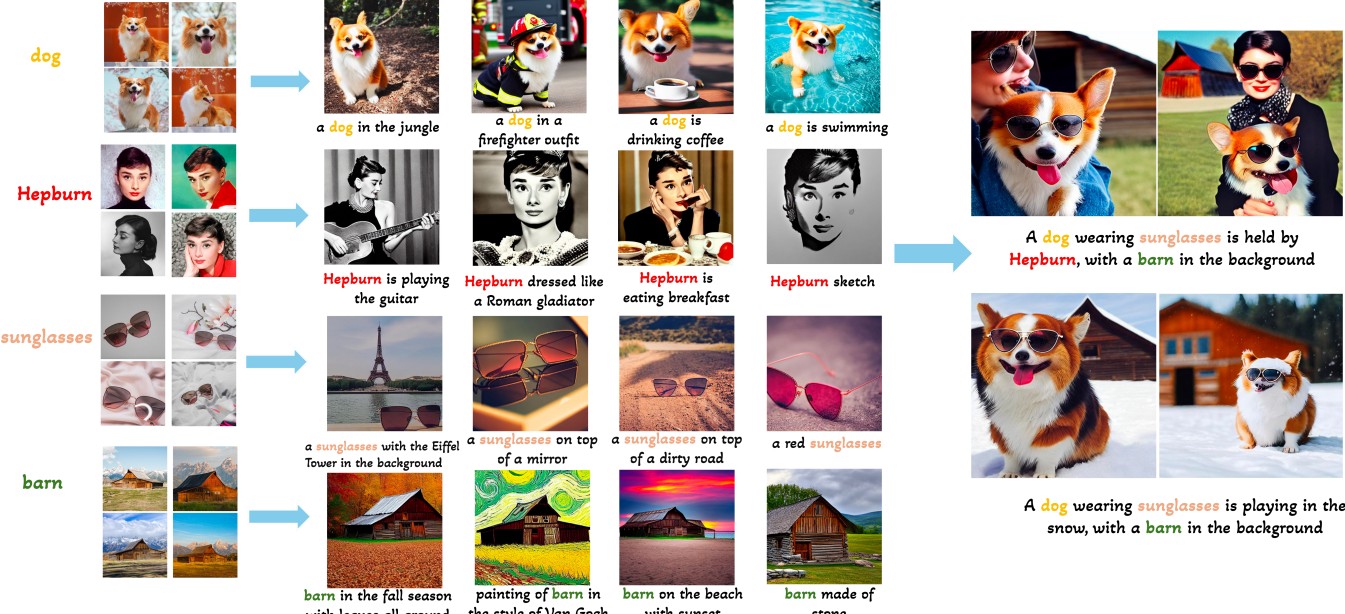

**Figure 1: Given a set of concept samples, Infusion demonstrates a remarkable ability to accurately assimilate these concepts and adeptly generate imaginative compositions guided by textual descriptions.**

*Corresponding Author.

*MM '24, October 28-November 1, 2024, Melbourne, VIC, Australia*

© 2024 Copyright held by the owner/author(s). Publication rights licensed to ACM.
ACM ISBN 979-8-4007-0686-8/24/10
https://doi.org/10.1145/3664647.3680894

## Abstract

Text-to-image (T2I) customization aims to create images that embody specific visual concepts delineated in textual descriptions. However, existing works still face a main challenge, **concept overfitting**. To tackle this challenge, we first analyze overfitting, categorizing it into concept-agnostic overfitting, which undermines non-customized concept knowledge, and concept-specific overfitting, which is confined to customize on limited diversities, *i.e*, backgrounds, layouts, styles. To evaluate the overfitting degree, we further introduce two metrics, *i.e*, Latent Fisher divergence and

Wasserstein metric to measure the distribution changes of non-customized and customized concept respectively. Drawing from the analysis, we propose Infusion, a T2I customization method that enables the learning of target concepts to avoid being constrained by limited training diversities, while preserving non-customized knowledge. Remarkably, Infusion achieves this feat with remarkable efficiency, requiring a mere **11KB** of trained parameters. Extensive experiments also demonstrate that our approach outperforms state-of-the-art methods in both single and multi-concept customized generation. Project page: https://zwl666666.github.io/infusion/.

## CCS Concepts

• **Computing methodologies → Computer vision**.

## Keywords

T2I customization, Overfitting, Concept-agnostic, Concept-specific

**ACM Reference Format:**
Weili Zeng, Yichao Yan, Qi Zhu, Zhuo Chen, Pengzhi Chu, Weiming Zhao, and Xiaokang Yang. 2024. Infusion: Preventing Customized Text-to-Image Diffusion from Overfitting. In *Proceedings of the 32nd ACM International Conference on Multimedia (MM '24), October 28-November 1, 2024, Melbourne, VIC, Australia.* ACM, New York, NY, USA, 10 pages. https://doi.org/10.1145/3664647.3680894

## 1 Introduction

In the era of multi-modality, text-to-image (T2I) generation [29, 31, 33, 35] has experienced rapid growth, showcasing highly imaginative works. Meanwhile, T2I customization rises as a potential need in image generation, aiming to create images that embody specific visual concepts aligned with textual descriptions. Users only need to provide a set of similar images whose pertinent visual concepts, *e.g*, pets, toys, or styles, are subsequently extracted by a T2I model. These identified concepts are then seamlessly interwoven into the users' textual descriptions for customized generation.

Typically, T2I customization methods can be classified into two categories: inversion-based and fine-tuning. Specifically, the inversion-based methods [11, 47] represent concepts by learning additional conceptual words. While this approach offers flexibility, it introduces a significant overfitting challenge, as the injected concepts permeate every part of the cross-attention module with textual information. Another approach, fine-tuning method [34], keeps the original architecture and optimizes model parameters with a set of customized image data. However, a large parameter space that is fitted by a small dataset leads to severe overfitting. Besides, the extensive size of these models often necessitates substantial storage and computation resources. Although Parameter-efficient tuning (PET) approaches [21, 24] shrink the fine-tuning into only a small subset of parameters, they still fail to eliminate the overfitting issue.

Comprehensively considering the previous methods, current T2I customization still faces three challenges: **1) Concept overfitting** takes away general generation capabilities of foundational T2I models from customized models. **2) Lack of appropriate metrics** to quantify the impact of overfitting on customized models. **3) Cumbersome deployment** prevents users from seamlessly switching between customized mode and regular mode as desired.

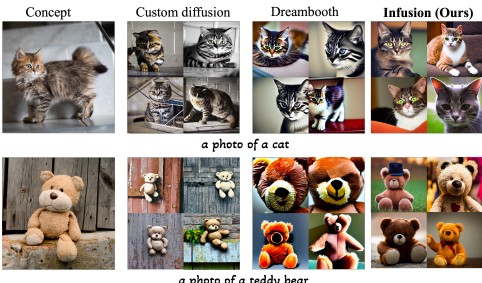

Concept  Custom diffusion  Dreambooth  **Infusion (Ours)**

*a photo of a cat*

*a photo of a teddy bear*

**Figure 2: Example of concept-agnostic overfitting. The first column on the left is the target concept, and the right is the non-customized results. The generated "cat" consistently exhibits black spotted stripes, while the "teddy" consistently presents a doll-like form, both sharing similar backgrounds.**

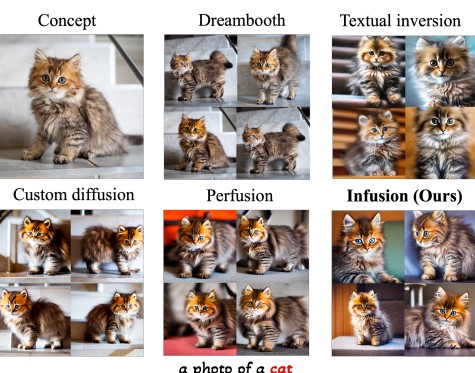

Concept  Dreambooth  Textual inversion

Custom diffusion  Perfusion  **Infusion (Ours)**

*a photo of a* **cat**

**Figure 3: Example of concept-specific overfitting. Customization with the prompt "a photo of a ⟨cat⟩", we reveal that all prior methods generate cat images with a similar size, pose, or background to the training data.**

We deeply explore the overfitting problem and categorize it into **concept-agnostic overfitting** and **concept-specific overfitting**. Concept-agnostic overfitting is depicted in Figure 2. When learning the concepts of ⟨cat⟩ and ⟨teddy⟩, customization undermines original generative ability for non-customized concept. Concept-specific overfitting is depicted in Figure 3. During the customization process of T2I model, it confuses target concept with limited training diversities, resulting in a diminution of generative diversity and consequently impacts the textual controllability.

Based on the analysis of the above two categories of overfitting, we introduce "Latent Fisher divergence" and "Wasserstein metric" to measure the distribution changes of non-customized and customized concept respectively. The latent Fisher divergence quantifies the score difference of the non-customized concept between the original model and the customized model. The Wasserstein metric quantifies the deviation of the customized distribution from its original generative distribution under a super-class concept, serving as a measure of diversity.

To overcome the aforementioned challenges of overfitting, we propose the Infusion method, a T2I customization method that enables the learning of target concepts to avoid being constrained

by limited training diversities. Infusion can also flexibly inject customized concepts into original models while preserving their original knowledge. Our novel insight lies in the decoupling of the attention map and value feature in each cross-attention module. As depicted in Figure 4, we implemented a dual-stream structure that shares pipeline parameters. The foundational T2I pipeline performs regular T2I generation. To harness its generative capabilities, we imitate the operations in P2P [17], and use the attention maps in the foundational T2I pipeline to replace the attention maps in the customized T2I pipeline. We introduce target concepts by learning a residual value embedding in the customized pipeline. All the learnable embeddings for a concept are highly lightweight, only 11KB, enabling seamless integration into the pipeline in a flexible and plug-and-play manner.

In summary, our contributions can be delineated as follows:

- We analyze concept-agnostic and concept-specific overfitting, and propose employing "Latent Fisher divergence" and "Wasserstein metric" to quantify their impact.
- We propose Infusion, a method that decouples attention maps and value features in each cross-attention module, fully utilizing the capability of the original T2I model.
- Experiments show that Infusion enables plug-and-play single-concept and multi-concept generation and achieves superior results compared to state-of-the-art methods.

## 2 Related Work

### 2.1 Text-to-image Generation

Early Text-to-Image (T2I) models primarily relied on Generative Adversarial Networks [6, 14, 22, 41, 51, 56], yet often exhibited limitations in terms of diversity. Recently, diffusion models [19, 38–40, 53, 54], coupled with classifier-free guidance [20], have emerged as leaders in this domain. Their remarkable generative performance is particularly noteworthy when applied to large-scale internet datasets [27, 31–33]. This fact drives us to fully exploit the prior knowledge of these foundational models for T2I customization.

### 2.2 Text-based Image Editing

The advent of contrastive multimodal models [30, 50], exemplified by CLIP [30], has introduced a transformative paradigm shift in the realm of image editing [3–5, 13, 28]. Significantly, these models empower global or localized image manipulation through the exclusive utilization of textual prompts. This paradigm has engendered a series of text-based image editing methods [7, 17, 23, 26, 52], among which the most relevant to our work is the prompt-to-prompt (P2P) [17]. P2P achieves image editing by replacing or modifying the attention map of the cross-attention module. Similarly, our approach employs attention map replacement for image customization. The difference is that P2P operates on virtual generated concepts, while we can operate on real concepts.

### 2.3 Text-to-image Customization

Text-to-Image (T2I) customization aims to generate customized concepts aligned with text description. Textual Inversion [11] is the pioneering method in this realm, which introduces an innovative approach to learning new word embeddings for representing specific concepts without tuning model parameters. Despite its plug-and-play adaptability, its efficacy remains constrained. DreamBooth [34] employs class nouns along with unique identifiers to represent target concepts but requires the comprehensive training of model parameters. While achieving high conceptual fidelity, this approach is susceptible to overfitting. Subsequent methodologies, incorporating low-rank updates or just tuning a few parameters, aim to address this limitation [15, 21, 24]. Notably, Custom Diffusion [24] focuses on updating the weights of cross-attention and introduces regularization sets to mitigate overfitting. Perfusion [42] introduces a decoupling of cross-attention into "what" and "where" pathways. It leverages a Rank-one Model editing method [25] to adjust the weights of key and value projection matrices, thereby alleviating concerns associated with overfitting. Another research trajectory involves using extensive customized data for pre-training encoders, enabling training-free inference [2, 10, 12, 48, 49, 55].

## 3 Analysis of Concept Overfitting

In this section, based on T2I diffusion model, we analyze two types of overfitting in the customized diffusion model: concept-agnostic overfitting and concept-specific overfitting. We subsequently define two metrics corresponding to each type of overfitting.

### 3.1 Preliminaries of T2I Diffusion Models

We applied our method to the Stable Diffusion (SD) model [1]. During the training phase, the encoder $\mathcal{E}$ maps the input image $x \in \mathcal{X}$ to latent code $z = \mathcal{E}(x)$. Subsequently, the decoder $\mathcal{D}$, is tasked with reconstructing the input image, aiming for $\mathcal{D}(\mathcal{E}(x)) \approx x$.

Diffusion models are then introduced to the latent space. The forward process perturbs the input to a noisy variant $z_t$, at each time step $t$. Given a conditional prompt $y$, the latent diffusion model $\varepsilon_\theta$ is to minimize the loss function:

$$\mathcal{L} = \mathbb{E}_{z \sim \mathcal{E}(x), y, \varepsilon \sim \mathcal{N}(0,1), t} \left[ \| \varepsilon - \varepsilon_\theta \left( z_t, t, \tau(y) \right) \|_2^2 \right], \quad (1)$$

where $\varepsilon_\theta$ is a time-conditional UNet [36] comprising self-attention layers and cross-attention layers. $\tau(y)$ is randomly replaced by an empty input $\varnothing$. This training objective can be construed as a form of denoising score matching [45], with $\varepsilon_\theta(z_t, t)$ serving as an approximation to the score $\nabla_z \log p(z_t)$ of the latent distribution.

Stable Diffusion employs a cross-attention mechanism to incorporate textual modal inputs. As previously outlined, the operation $\tau_\theta$ projects the text condition $y$ into an intermediate representation denoted as $\tau_\theta(y) \in \mathbb{R}^{L \times d_\tau}$. Subsequently, the cross-attention operation of the $i$-th layer network is expressed as:

$$Atten(Q_i, K_i, V_i) = \text{softmax} \left( \frac{Q_i K_i^T}{\sqrt{d}} \right) V_i = M_i V_i, \quad (2)$$

which is utilized to integrate textual information into the UNet network. The constituents are defined as:

$$Q_i = \varphi_i \left( f_i \right) \cdot W_Q^i, K_i = \tau_\theta(y) \cdot W_K^i, V_i = \tau_\theta(y) \cdot W_V^i, \quad (3)$$

where $Q_i \in \mathbb{R}^{N_i^2 \times d}$, $K_i \in \mathbb{R}^{L \times d}$, and $V_i \in \mathbb{R}^{L \times d}$ respectively represent the features of the query, key, and value. $\varphi_i \left( f_i \right) \in \mathbb{R}^{N_i^2 \times d_\epsilon^i}$ represents the (flattened) latent image representation of the UNet, and $W_Q^i \in \mathbb{R}^{d_\epsilon^i \times d}$, $W_K^i \in \mathbb{R}^{d_\tau \times d}$, and $W_V^i \in \mathbb{R}^{d_\tau \times d}$ are learnable matrix parameters [33].

Weili Zeng, Yichao Yan, Qi Zhu, Zhuo Chen, Pengzhi Chu, Weiming Zhao, and Xiaokang Yang

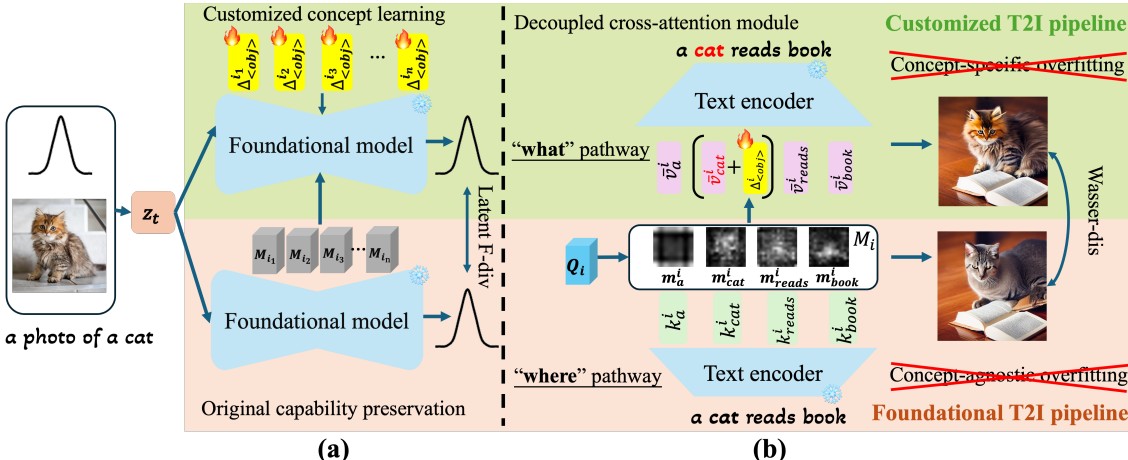

**Figure 4: Infusion pipline. (a) Infusion fully preserves the capacity of the original model, precluding concept-agnostic overfitting. (b) Infusion decouples the cross-attention module, precluding concept-specific overfitting.**

## 3.2 Analysis of Concept-agnostic Overfitting

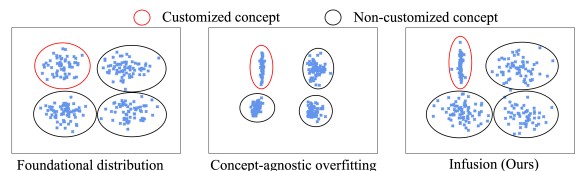

**Figure 5: Concept-agnostic overfitting. Customized tuning, as observed in Dreambooth [34], undermines non-customized generative capabilities.**

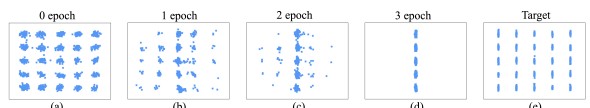

**Figure 6: Concept-specific overfitting. The confusion training between customized concepts and limited diversities gradually reduces the number of original diversities.**

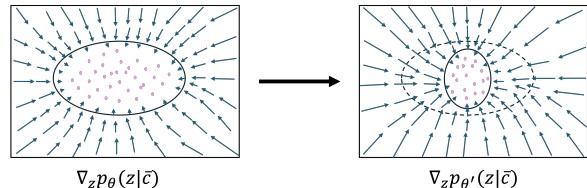

**Figure 7: Shrinking of non-customized distribution. The gradient field $\nabla_z \log p_\theta(z|\bar{c})$ gradually contracting during training, leading to a more convergent field $\nabla_z \log p_{\theta'}(z|\bar{c})$.**

**Concept-agnostic overfitting** often stems from alterations in the foundational model parameters aimed at customization, thereby compromising the model's performance on non-customized concepts. As shown in Figure 5, our toy foundational model is a four-peak hybrid Gaussian model, where each peak represents a different concept. The process of customized tuning can be viewed as the model learning a specialized distribution under a particular concept. Here, we simplify it as learning a linear distribution under the Gaussian peak in the top-left corner. It can be observed that when employing a tuning method similar to Dreambooth [34], the generated distributions for non-customized concepts become compact. The shrinking of these distributions implies a reduction in the diversity of generated samples under non-customized concepts.

Therefore, for T2I models, we infer that concept-agnostic overfitting is primarily related to the offset of the non-customized score. Specifically, let $c$ represent the customized concept and $\bar{c}$ represent non-customized concepts. $\theta$, $\theta'$ are the parameters of the original model and customized model. As depicted in Figure 7, the gradient field described by $\nabla_z \log p_\theta(z|\bar{c})$, gradually contracting during training, leading to a more convergent field $\nabla_z \log p_{\theta'}(z|\bar{c})$. To quantify this effect, we define the latent Fisher divergence as definition 3.1, which measures the distance between two models' score under the non-customized concepts.

**Definition 3.1** (Latent Fisher divergence). Based on the fact that $c$ is only a small part of a vast image concept, and the training principles of denoise score matching [45], the Fisher divergence between $p_\theta(z|\bar{c})$ and $p_{\theta'}(z|\bar{c})$ can be approximated by:

$$
\begin{aligned}
&D_F\left(p_\theta(z|\bar{c})\|p_{\theta'}(z|\bar{c})\right) \\
&= \frac{1}{2}\mathbb{E}_{z\sim\mathcal{E}(x)}\left[\|\nabla_z \log p_\theta(z|\bar{c}) - \nabla_z \log p_{\theta'}(z|\bar{c})\|_2^2\right] \\
&\approx \frac{1}{2}\mathbb{E}_{z\sim\mathcal{E}(x)}\left[\left\|\varepsilon_{\theta(z_t,t,\bar{c})} - \varepsilon_{\theta'(z_t,t,\bar{c})}\right\|_2^2\right].
\end{aligned}
\tag{4}
$$

In essence, this metric assesses the degree to which the model's inherent generative capacity is compromised throughout the customization process.

## 3.3 Analysis of Concept-specific Overfitting

**Concept-specific overfitting** primarily stems from the confusion learning between customized concepts and limited training diversities. As illustrated in Figure 6, we also employ a toy Gaussian model for clarification, where each peak region represents different diversities within the same concept. The original foundational model is capable of generating 25 different diversities within a similar concept, each representing variations such as diverse backgrounds, layouts, and styles. The customized target is also to learning a linear distribution. Typically, customized training datasets only contain a subset of these diversities. We represent this with the utilization of only five Gaussian distributions. It is evident from Figure 6 that, during the training process, the model gradually reduces the number of diversities and ultimately converges to only include those present in the training dataset. However, as shown Figure 6(e), our ultimate goal is to represent linear concepts as comprehensively as possible across all diversities within the model.

Therefore, during the learning process of T2I model, we infer that the model confuses the target concept with other diversities such as background, layout, style, etc., leading to concept-specific overfitting. Based on the manifold hypothesis [16], each concept's generative distribution can be regarded as a probability manifold embedded within the entire generative space, with the distributions under different diversities of each concept considered as distinct sub-manifolds embedded within this manifold. To quantify this overfitting, it becomes necessary to measure the distance between these sub-manifolds before and after customization training. To achieve this, we introduce the Wasserstein metric, as defined in definition 3.2.

**Definition 3.2** (2-Wasserstein metric [44]). Let $(\mathcal{X}, d)$ be a Polish metric space. For any two probability measures $\mu, \nu$ on $\mathcal{X}$, $\mu$ and $\nu$ are respective marginal distributions over $\mathcal{X}$, law $(X) = \mu$, law $(Y) = \nu$. $d(\mathbf{x}, \mathbf{y}) = \|\mathbf{x} - \mathbf{y}\|_2$ is Euclidean distance. $\pi(x, y)$ is the set of all probabilistic couplings in $(\mathcal{X} \times \mathcal{X})$ with marginals $\mu$ and $\nu$. The 2-Wasserstein distance between $\mu$ and $\nu$ is defined by the formula:

$$W_2(\mu, \nu) = \left( \inf_{\pi \in \Pi(\mu, \nu)} \int_{\mathcal{X}} d(x, y)^2 d\pi(x, y) \right)^{1/2}$$
$$= \inf \left\{ \left[ \mathbb{E}d(X, Y)^2 \right]^{\frac{1}{2}} \right\}. \tag{5}$$

However, directly compute Wasserstein metric in high-dimensional space is difficult. So in this work, inspired by Fréchet Inception Distance [18], we fits a Gaussian distribution to the latent of SD model for each distribution and then computes the 2-Wasserstein distance, between those Gaussians.

## 4 Method

To address the aforementioned two types of overfitting, we aim to retain the generative capacity of the foundational T2I model and exploit its generative diversity for customization. For instance, to generate a customized image with the prompt "a ⟨cat⟩ is reading a book". An intuitive strategy is that we generate an initial image by foundational T2I model first. During this process, we can apply a transformation to replace the depicted "cat" with the desired customized "⟨cat⟩". Note that the use of "⟨⟩" is merely for written distinction, and we do not undertake additional learning of unique

identifiers. As shown in Figure 4, the "cat" was selectively replaced with a customized concept, while preserving the constancy of other elements. In this way, we retain the T2I model's generative capacity, avoid concept-agnostic overfitting, and simultaneously harness its diversity to mitigate the risk of concept-specific overfitting.

## 4.1 Cross-attention Affects T2I Generation

Following [42], we can partition the cross-attention module into two pathways: the "where" pathway for the position and geometric attributes of objects in the final image, and the "what" pathway dictating the features incorporated into each spatial region, thus controls the content appearing in the final image. Specifically, expanding the expression denoted by Equation 2, we derive the following:

$$Atten(Q_i, K_i, V_i)$$
$$= \left( m_1^i, m_2^i, \cdots, m_L^i \right) \left( (v_1^i)^\top, (v_2^i)^\top, \cdots, (v_L^i)^\top \right)^\top$$
$$= \sum_{k=1}^{L} m_k^i \cdot v_k^i. \tag{6}$$

Here, $m_k^i \in \mathbb{R}^{N_i^2 \times 1}$ denotes the attention map corresponding to the $k$-th token of the input prompt and $v_k^i \in \mathbb{R}^{1 \times d}$ represents the value feature associated with the same token. As shown in Figure 4, there is usually minimal overlap between attention maps of the foundational T2I model. Various methods [9, 43] exist can be directly combined with Infusion to further prevent their overlap, here we just consider a simplified scenario. Thus, in conjunction with Equation 6, we observe that independent attention maps determine the positional presentation of their corresponding tokens in the image. The content they present is determined by the respective values.

Under the condition where the conceptualized generation content remains fixed, the augmentation of spatial layout and posture in the generated entities is imperative to amplify the diversity of customized generation. Furthermore, we posit that the foundational T2I model, untainted by customized training, inherently encapsulates the spatial diversity. To fully harness the diversity, our insight is that its attention maps should be decoupled for customized generation.

It is somewhat akin to the mechanisms employed in P2P, we extract attention maps from the foundational T2I model's generation under identical prompts, utilizing them to replace the attention maps in the customized generation process. However, P2P is constrained to the substitution of concepts exclusively within the generative scope of the model.

## 4.2 Concept Learning with Residual Embedding

The essence of Infusion lies in preserving the cross-attention maps of the foundational T2I model and subsequently learning and infusing concepts during each time step $t$ along the "what" pathway.

For example, consider the training image $I$ and its corresponding text description $y$ ("a photo of a cat"). We feed them into both the foundational T2I pipeline (F-pipeline) and the customized T2I pipeline (C-pipeline). These two pipelines share fixed foundational

model parameters. Differently, to facilitate the learning of customized concepts, a trainable embedding is introduced in the "what" pathway of the C-pipeline. Specifically, a residual embedding $\Delta_i^{\text{cat}}$ is incorporated into the value feature of the "cat" token. This embedding represents the customized concept and fulfills the transformation from the original category to a specific object. Consequently, the attention module within the C-pipeline can be structured as follows:

$$
\begin{aligned}
\overline{Atten}&(Q_i, K_i, \overline{V_i}) \\
&= \sum_{k \neq \text{obj}} m_k^i \cdot \overline{v_k^i} + m_{\text{obj}}^i \cdot \overline{v_{\text{obj}}^i} \\
&= \sum_{k \neq \text{obj}} m_k^i \cdot v_k^i + m_{\text{obj}}^i \cdot (v_{\text{obj}}^i + \Delta_{\langle \text{obj} \rangle}^i).
\end{aligned}
\tag{7}
$$

The subscript $obj$ denotes the index of the concept token in the prompt. Vectors with "−" denote those from the C-pipeline, while vectors without "−" originate from the F-pipeline. It is noteworthy that $\overline{v_k^i} = v_k^i$ when $k \neq \text{obj}$, or in other words, $\overline{V}$ differs from $V$ only in the value feature corresponding to the token $\langle \text{obj} \rangle$.

Finally, the trainable parameters $\Delta_{\langle \text{obj} \rangle}^i$ are optimized with the customization loss without additional regularization:

$$
\mathcal{L} = \mathbb{E}_{z \sim \mathcal{E}(x), y, \varepsilon \sim \mathcal{N}(0,1), t} \left[ \left\| \varepsilon - \varepsilon_{\theta'} \left( z_t, t, \tau(y) \right) \right\|_2^2 \right],
\tag{8}
$$

where $x$ is the reference few-shot data of $\langle \text{obj} \rangle$, $\theta'$ is the parameters of the customized model.

### 4.3 Single-concept and Multi-concept Inference

Similarly, during the inference stage, attention maps from the F-pipeline are employed to replace the attention maps within each cross-attention module in the C-pipeline. The previously obtained residual value embedding can be effortlessly applied to value features of the corresponding concept token. For a prompt involving $S$ specific concepts $\langle \text{obj}_1 \rangle, \langle \text{obj}_2 \rangle, \cdots, \langle \text{obj}_S \rangle$, it is sufficient to retrieve the pre-computed $\Delta_{\langle \text{obj}_1 \rangle}^i, \Delta_{\langle \text{obj}_2 \rangle}^i, \cdots, \Delta_{\langle \text{obj}_S \rangle}^i$ and integrate them into the value embedding of the $i$-th layer, as follows:

$$
\overline{v_{\text{obj}_s}^i} = v_{\text{obj}_s}^i + \Delta_{\langle \text{obj}_s \rangle}^i.
\tag{9}
$$

Combining Equation 7 with Equation 9, we can finally generate single-concept or multi-concept customized images.

sectionExperiments

### 4.4 Experiment Details

**Experiment Setup.** We utilize the pre-trained StableDiffusion (SD)-v1.5 model [1] for ablation studies and comparisons. In each layer of the Unet's cross-attention, we train the residual concept $\Delta_{\langle \text{obj} \rangle} \in \mathbb{R}^{1 \times d}$ for the corresponding value feature, where $d = 768$. The optimization of these embeddings is carried out with a learning rate of 0.01 and a batch size of 4. During the inference process, we employ a DDIM sampler with a sampling step size of $T = 50$, and leverage classifier-free guidance with a guiding scale of $s = 8$. We utilize datasets from prior works [24, 34], encompassing diverse subjects such as toys, animals, buildings, and personal items.

**Baseline.** We benchmark our method against state-of-the-art competitors, including Textual Inversion, DreamBooth, DreamBooth-lora, Custom Diffusion, and Perfusion. For Textual Inversion and DreamBooth(lora), we utilize their Diffusers versions [46], and for Custom Diffusion and Perfusion, we employ their official implementations with experimental parameters configured following official recommendations.

**Overfitting Evaluation.** Referring to Section 3, we employ the Latent Fisher divergence and Wasserstein metric to assess the concept-agnostic overfitting and concept-specific overfitting on the customized model. Latent Fisher divergence is computed based on a random sample of 50000 examples from LAION-400M [37]. We sample time steps uniformly and measure the distance between the outputs of $\epsilon_\theta$ and $\epsilon_{\theta'}$. To evaluate concept-specific overfitting, we generate 1000 samples under 8 concepts based on 20 prompts and calculate the 2-Wasserstein distance between $p_\theta(z|x, c)$ and $p_{\theta'}(z|x, c)$.

**Customization Evaluation.** Following previous works [34, 48], we employ CLIP-I, CLIP-T, and DINO-I to evaluate the model ability for customized image generation. CLIP-T is to assess the alignment between generated images and prompts. CLIP-I evaluates the similarity between generated samples and training data in the CLIP feature space. DINO-I computes the cosine similarity between the ViTS/16 DINO [8] embeddings of training images and generated images.

### 4.5 Overfitting Comparison

To ensure fair comparison of the anti-overfitting capabilities among different customized models, we adjusted the learning rate and batch size to achieve near-optimal performance for each method at the same number of training steps. Additionally, we continued training each model after convergence to assess the extent of overfitting. As depicted in Figures 10 and Figures 11, Infusion demonstrates robust resistance to concept overfitting compared to other methods. As illustrated in Figure 12, we present the generated results of Infusion at training steps 100, 200, 400, 1000 and 2000. It is evident that even at higher training steps, Infusion keeps producing content highly aligned with the given text. This sets it apart from other methods that require precise control over training steps to prevent overfitting. In contrast, DB(Dreambooth)-lora, CD(Customized Diffusion), and TI(Textual Inversion) exhibit increasing susceptibility to concept-specific overfitting with continued training, becoming increasingly confined to the training diversities, and losing control over textual consistency. Perfusion performs better in this regard, but its consistency with customized concepts is notably inferior to Infusion. Moreover, since Infusion can directly generate non-customized concepts through the F-pipeline, it remains unaffected by concept-agnostic overfitting.

### 4.6 Qualitative Results

Figure 8 illustrates a series of generated images produced by our approach and its competitors, including Stable Diffusion (SD), for single-concept customization. To rigorously assess the models' generative capabilities, we deliberately selected a set of imaginative descriptive prompts. Notably, DreamBooth and Textual Inversion

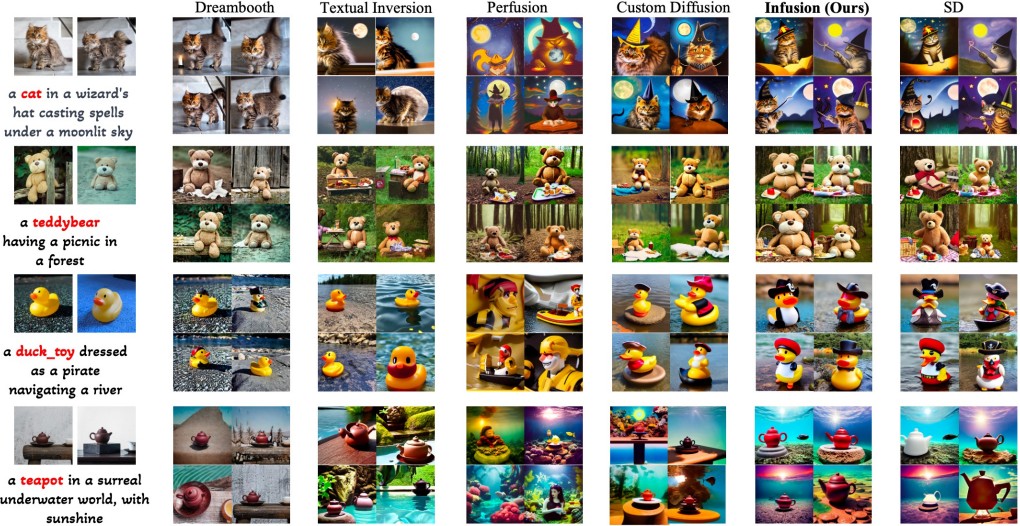

**Figure 8: Single-concept generation. The visual comparison involves multiple methods, with Infusion demonstrating robust customization capabilities to align textually and conceptually.**

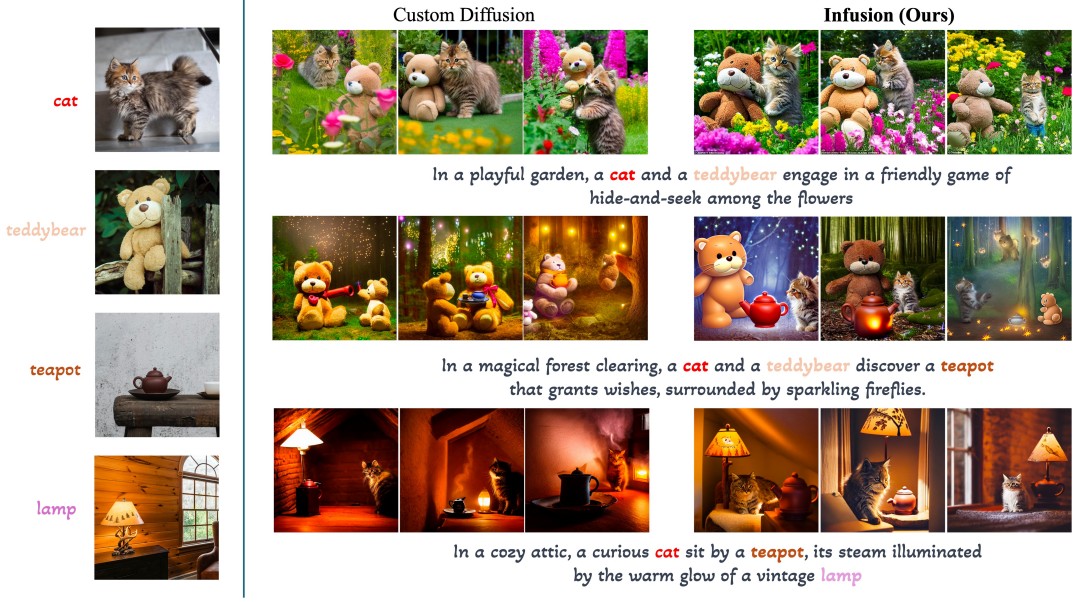

**Figure 9: Multi-concept generation. We show the results of multi-concept. Infusion demonstrates superior performance in generating works that exhibit higher levels of imagination and fidelity to customized concepts compared to Custom Diffusion**

exhibit pronounced susceptibility to overfitting, emphasizing conceptual consistency limits their imagination. Perfusion, on the other hand, demonstrates commendable imaginative capacities, yet tends to overlook the generation of customized concepts. Custom Diffusion, while achieving performance comparable to our method, still lacks some diversity.

In contrast, Infusion adeptly balances textual expression and concept fidelity. Furthermore, when compared with the generated

results of SD, it becomes evident how our method leverages the "where" pathway and "what" pathway to infuse concepts. Our approach effectively preserves the SD's generated background and layout while injecting customized concepts onto the corresponding objects. This strategy allows us to harness the imaginative capacity of SD for customization and avoid overfitting.

Figure 9 showcases the performance of our method in multi-concept generation. In this context, we only present the results

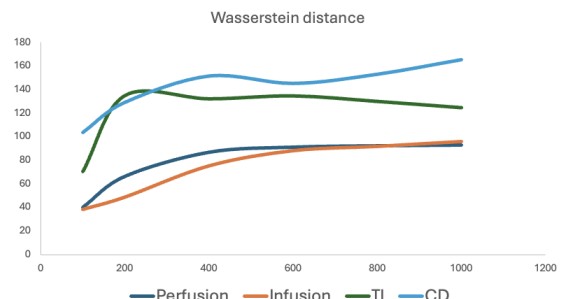

**Figure 10: Wasserstein metric at various training steps.**

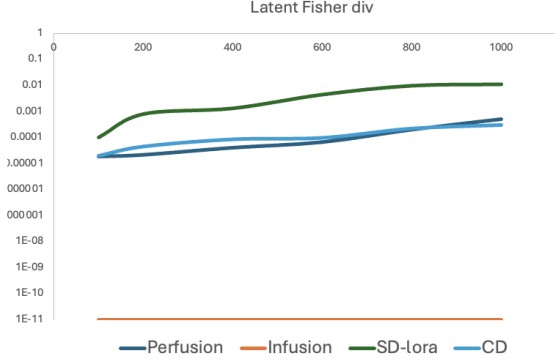

**Figure 11: Latent Fisher Divergence at various training steps.**

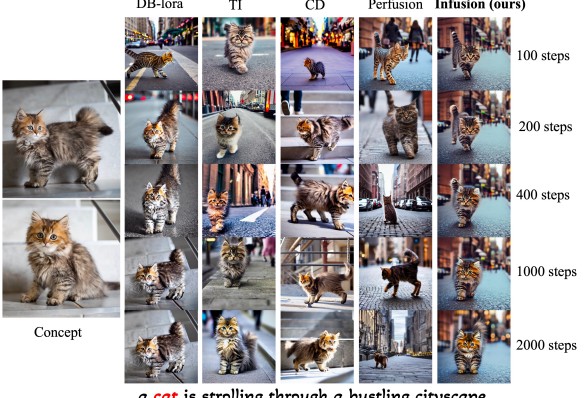

**Figure 12: Customized generation results at various training steps.**

of Infusion and Custom Diffusion, as the performance of other competitors in this task significantly lags behind these two. We observe that Custom Diffusion tends to overlook some concepts during multi-concept customization. In comparison, our method excels in crafting works that are not only more imaginative but also more faithful to the customized concepts.

## 4.7 Quantitative Results

**Table 1: Quantitative Results. We compared different baseline methods across various metrics, and Infusion achieved the highest text alignment score.**

| Method | CLIP-T ↑ | CLIP-I ↑ | DINO ↑ | Storage |
|---|---|---|---|---|
| DreamBooth [34] | 0.255 | **0.838** | **0.721** | 3.3 GB |
| DreamBooth-lora [46] | 0.264 | 0.791 | 0.615 | 3.29 MB |
| Textual Inversion [11] | 0.240 | 0.780 | 0.584 | **3**KB |
| Custom Diffusion [24] | 0.256 | 0.800 | 0.643 | 57.1MB |
| Perfusion [42] | 0.242 | 0.729 | 0.505 | 100KB |
| **Infusion (Ours)** | **0.267** | 0.816 | 0.697 | 11KB |

Our quantitative analysis primarily focuses on general objects customization. Utilizing 20 concept dataset from [24, 34] and the same 25 prompts as in [34], we generated 5 images for each prompt. Table 1 demonstrates our method's superiority in text alignment. While it may not surpass Dreambooth and Custom Diffusion in image alignment and DINO score, we attribute this discrepancy to inherent limitations in the metrics. These metrics consider background information similarity during computation, which may not align with the goals of our customized generation approach. The simplicity of testing prompts also limits a comprehensive assessment, future work will include more intricate test texts to evaluate the model's generative capabilities thoroughly.

**Table 2: Quantitative Results. In each paired comparison, Infusion is preferred (over 50%) over the baseline methods in both text- and image-alignment.**

| vs Method | Text Alignment ↑ | Image Alignment ↑ |
|---|---|---|
| DreamBooth | 86.90% | 51.19% |
| Textual Inversion | 82.14% | 82.14% |
| Custom Diffusion | 69.05% | 75.00% |
| Perfusion | 65.48% | 84.52% |

**User Study**. We evaluate the average preferences of 21 participants for Infusion and other baseline methods. Each participant responds to 32 questions about quality comparison. Table 2 indicates users' general preference for Infusion in terms of text alignment and conceptual fidelity.

## 5 Conclusion and Limitations

In this work, we analyze two types of overfitting and introduce "Latent Fisher divergence" and "Wasserstein metric" to quantify them. Our proposed Infusion preserves the inherent generative capacity of the original T2I model while offering flexible, plug-and-play usage for customization. Infusion demonstrates excellent performance in overall learning of target concepts. However, for tasks that require high-fidelity preservation of detailed textures, Infusion exhibits certain limitations. In such cases, there is a trade-off between diversity and fidelity, necessitating the sacrifice of some diversity in exchange for higher fidelity, such as training more residual embeddings.

# Acknowledgments

This work was supported in part by NSFC (62201342, 62101325), and Shanghai Municipal Science and Technology Major Project (2021SHZDZX0102).

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

Received 20 April 2024; revised 1 July 2024; accepted 18 July 2024