# OpenReview forum: "Infusion: Preventing Customized Text-to-Image Diffusion from Overfitting"
_acmmm.org/ACMMM/2024/Conference — MM2024 Poster_

### Official Review · Reviewer_4wNo · 2024-05-09

**Rating:** 4
**Confidence:** 4

**Summary:**

This paper conducts a detailed analysis of the overfitting problem that occurs in customized finetuning. The authors interpret the problem theoretically and claim that the overfitting problem mainly consists of concept-agnostic overfitting and concept-specific overfitting. This paper proposes “Latent Fisher divergence” and “Wasserstein metric” to quantify them, and the experimental results show the visually superior performance.

**Strengths:**

1. Experimental results show the better performance than other costomized finetuning methods, e.g., dreambooth, textual inversion, custom diffusion, etc.
2. The proposed method achieves this feat with remarkable efficiency, requiring a mere 11KB of trained parameters.
3. The paper is well organized and analyzes the problem both theoretically and experimentally.

**Limitations:**

After reading the paper, I have some doubts as follows:
1. In fig.2 and fig.3, why does the proposed method (Infusion) have such differentiated output under the same prompt (a photo of cat)? In fig.2, "a photo of cat" corresponds to diverse cat images, while in fig.3, "a photo of cat" corresponds to output images of the same cat category.
2. In fig.8, in the images generated by Infusion, why is the overall brightness of the object much brighter compared to the concept images, especially in the example of teapot.
3. This paper introduces “Latent Fisher divergence” and “Wasserstein metric”, but lacks an ablation study on these two. I wonder which core technology is responsible for the performance improvement of this method.
4. Generally speaking, how many concept images does this method require to achieve the effect of generating realistic images? What's the relationship between the performance and the number of the concept images?

**Suitability:**

3

---

### Official Review · Reviewer_Wh2t · 2024-05-24

**Rating:** 3
**Confidence:** 3

**Summary:**

This paper analyzes the overfitting issue in text-to-image customization tasks, classifying it into two categories based on whether the overfitting phenomenon is related to the specific concept needing customization. It introduces two metrics to measure these overfitting categories during text-to-image customization. Furthermore, the paper proposes a model named Infusion to address these issues.

**Strengths:**

1. Compared to existing state-of-the-art methods, the proposed method demonstrates performance improvement according to the presented generated images and it is simple but effective.

2. The paper provides an in-depth exploration of the overfitting issue in text-to-image customization and visualizes it through two simple experiments, presenting the issue clearly.

**Limitations:**

1. The terminology used in the paper is confusing, particularly the term "training modalities," which appears frequently. It is inappropriate to use this phrase to describe the diversity of training data since modalities typically refer to data types such as video, image, text, depth, etc.

2. Several symbols in the paper lack proper explanations, making the equations difficult to understand. For example, the symbols in equation (5) in L486-L490 are not adequately explained.

3. There are some unclear descriptions in the paper. For instance, in L272-L274, it mentions “the difference is that P2P ...”, but the concept of the "virtual generated concept" in this sentence is confusing and lacks further explanation. Additionally, P2P can also be applied to edit real images through an existing inversion process.

4. There are issues with the proposed two metrics and the analysis of overfitting in S3:

(1) The paper uses two simplified experiments to imitate text-to-image customization with diffusion models. The scientific basis for this simplification is unclear.

(2) The metrics for concept-agnostic and concept-specific overfitting are both based on calculating the difference between two distributions. It is unclear why they are not derived from the same metric for distribution distance measurement.

(3) In L487-L479, it mentions, “To quantify ...”, but the method for obtaining the distribution of the target concept is unclear. The target concept cannot be sampled from the foundational distribution of the pre-trained diffusion model, and it is impossible to present all images with the target concept. In this paper, this distribution is estimated by sampling data with generalized text related to the target concept in the foundational distribution, which is confusing.

5. There are issues with the experiments in S5.2. The analysis of Figures 10 and 11 is not appropriate. For example, the phrases "excellent fidelity to textual consistency" in L670-L672 and “losing control ..." in L676 are inappropriate. Figures 10 and 11 only reflect the overfitting condition of the model and do not demonstrate the consistency between the text prompts and generated images.

**Suitability:**

3

---

### Official Review · Reviewer_QQKJ · 2024-05-24

**Rating:** 4
**Confidence:** 3

**Summary:**

This paper introduces a method called Infusion aimed at mitigating overfitting in text-to- image (T2I) diffusion models. The authors identify two primary types of overfitting: concept-agnostic overfitting, which affects non-customized concepts, and concept- specific overfitting, which limits the diversity of generated images. To address these issues, Infusion employs a dual-stream structure that decouples the attention maps and value features in each cross-attention module. This approach leverages the foundational model's generative capacity while introducing customized concepts through a lightweight embedding process. The method uses metrics like Latent Fisher divergence and the Wasserstein metric to evaluate overfitting and demonstrates superior performance compared to state-of-the-art methods through extensive experiments.

**Strengths:**

1. Adequate analysis: The paper conducts adequate analysis on two types of overfitting in the customized diffusion model: concept-agnostic overfitting and concept-specific overfitting. And it define two metrics corresponding to each type of overfitting.
2. Novelty: Infusion decouples the attention mapping and value features, thereby leveraging the diverse generation modalities of the original T2I model to mitigate concept-specific overfitting
3. Adequate Evaluation:The paper provides comprehensive evaluations using both quantitative metrics and qualitative assessments. Extensive comparisons with state-of-the-art methods and user studies bolster the credibility of the results. The use of metrics tailored to measure overfitting is particularly noteworthy.

**Limitations:**

1. Application: The paper somehow solve the overfitting problem, but will it have bad influence on the image quality? As shown in Fig 8, the visualization of Infusion is not as good as dreambooth in the third row in the upper right: the duck toy losses its face. The author is suggested to show the quantitive comparison of the fid and clip score compared with existing methods to evaluate the t2i quality.
2. Multi-concept results: all the images in the figure9 in third cow has some unpleasant airtifacts, (such as the face of the cat in the first row, the light in a strange cat shape in the air in the second line and the lamp head with the teapot base), I wonder it doesn't really work in practical applications.
3. Poor Writing: The paper seems to written in a rush, there are obvious typos and organization errors. Such as:
a. line 93-97 "Generate the Correct Terms for Your Paper." b. Line 202 :"As c in the Figure"
c. the line 397-400 shouldn't be included in section 3.2 cause it also introduce the Concept-specific Overfitting.

**Suitability:**

3

---

### Meta-Review · Area_Chair_jLej · 2024-07-01

**Recommendation:** Accept (Poster)
**Confidence:** 5

**Metareview:**

This paper ultimately received three "borderline accept" scores. All reviewers were satisfied with the "novelty and positive results" of the paper. After the rebuttal, reviewers QQKJ and 4wNo felt that their questions were well addressed. Reviewer Wh2t thinks that "the description and the analysis" still need further improvement. Based on the above, I recommend accepting this paper, and strongly suggest that the authors further improve the paper according to Reviewer Wh2t's feedback.